# High-Throughput 16S rRNA Sequencing to Assess Potentially Active Bacteria and Foodborne Pathogens: A Case Example in Ready-to-Eat Food

**DOI:** 10.3390/foods8100480

**Published:** 2019-10-11

**Authors:** Marina Mira Miralles, Lucia Maestre-Carballa, Monica Lluesma-Gomez, Manuel Martinez-Garcia

**Affiliations:** Department of Physiology, Genetics and Microbiology, University of Alicante, Building 12, C/San Vicente, 03080 Alicante, Spain; marinamm2704@gmail.com (M.M.M.); lumc46@gmail.com (L.M.-C.); m.lluesma@ua.es (M.L.-G.)

**Keywords:** ready-to-eat salads, vegetable, lettuce, foodborne, pathogen, bacteria, 16S rRNA gene, next-generation sequencing, active bacteria

## Abstract

Technologies to detect the entire bacterial diversity spectra and foodborne pathogens in food represent a fundamental advantage in the control of foodborne illness. Here, we applied high-throughput 16S rRNA sequencing of amplicons obtained by PCR and RT-PCR from extracted DNA and RNA targeting the entire bacterial community and the active bacterial fraction present in some of the most consumed and distributed ready-to-eat (RTE) salad brands in Europe. Customer demands for RTE food are increasing worldwide along with the number of associated foodborne illness and outbreaks. The total aerobic bacterial count in the analyzed samples was in the range of 2–4 × 10^6^ CFU/g (SD ± 1.54 × 10^6^). Culture validated methods did not detect *Salmonella* spp., *Escherichia coli*, and other fecal coliforms. 16S rRNA gene Illumina next-generation sequencing (NGS) data were congruent with these culture-based results and confirmed that these and other well-known foodborne bacterial pathogens, such as *Listeria*, were not detected. However, the fine-resolution of the NGS method unveiled the presence of the opportunistic pathogens *Aeromonas hydrophyla* and *Rahnella aquatilis* (relative frequency of 1.33–7.33%) that were metabolically active in addition to non-pathogenic, active members of *Yersinia* spp. (relative frequency of 0.0015–0.003%). The common ail and foxA marker genes of *Yersinia enterocolitica* were not detected by qPCR. Finally, our NGS data identified to non-pathogenic *Pseudomonas* spp. as the most abundant and metabolically active bacteria in the analyzed RTE salads (53–75% of bacterial abundance). Our data demonstrate the power of sequencing, in parallel, both 16S rRNA and rDNA to identify and discriminate those potentially and metabolically active bacteria and pathogens to provide a more complete view that facilitates the control of foodborne diseases, although further work should be conducted to determine the sensitivity of this method for targeting bacteria

## 1. Introduction

Ready-to-eat (RTE) raw leafy vegetable-based salads are becoming very popular and widely accepted in our daily diet, and in recent years, these products have clearly covered and satisfied a general growing demand of consumers to incorporate “healthy and green” food. Good practices of hygiene at harvest, such as water quality, and postharvest involving fresh-cutting, washing—with or without sanitizers—and packaging is of central importance to ensure both public health protection and product quality since, in each one of these steps, cross-contamination with human pathogens might occur [1] Unfortunately, parallel to the popularity of this product, the number of outbreaks and cases of foodborne illness associated with the consumption of RTE salads is increasing [1]. Among a wide range of pathogens causing foodborne illnesses, *Escherichia coli* O157:H7, *Salmonella* spp., and *Listeria monocytogenes* are the most common pathogens that contaminate RTE salads [2]. Although official outbreaks linked to RTE salads have not been reported in Spain, in other countries, such as USA, two recent multistate outbreaks of *E. coli* O157:H7 infections linked to fresh lettuce were confirmed, with 272 people infected, 120 hospitalized, and 5 dead since June 2018 (official reports from Centres for Disease Control, EEUU). Unfortunately, this is only one example, and many more have been reported in other countries [3]. For instance, in UK, a national Salmonellosis outbreak was reported [4], and in Switzerland, foodborne transmission of *L. monocytogenes* was confirmed during 2013–2014 [5]. The latest official European Union report on foodborne outbreaks in 2017 confirmed that 4.2% of analyzed RTE salad samples contained *L. monocytogenes*. In addition to foodborne bacterial pathogens, norovirus has also been the focus of investigations related to RTE salad outbreaks [3,6,7]. Finally, although less attention has been paid, the presence of different protozoans among *Giardia duodenalis*, *Cryptosporidium* spp., *Toxoplasma gondii*, and *Cyclospora cayetanensis* has been confirmed in a large-scale study on RTE salads, with a prevalence of ≈4% of the analyzed samples [8].

The technologies to study and detect foodborne pathogens have evolved significantly over the last decade, but essentially in national health systems, most reference standards and validated approaches are based on the culture isolation of suspicious bacterial pathogens [9]. Although these methods are undoubtedly useful, they are biased according to the specific culture requirements for most genera and species and in some cases require long incubation periods [10]. In turn, methods that, in the same assay, can theoretically detect and target the entire bacterial diversity spectra regardless of the type of bacteria and pathogens present in a food sample represent a fundamental advantage in the analysis and control of foodborne illness. Nowadays, there are two nucleic acid-based techniques that allow one to characterize and target the bacterial community: 16S rRNA gene amplicon sequencing and metagenomics [11,12,13]. Both approaches allow to detect, identify, and monitor foodborne bacteria [14]. High-throughput sequencing of 16S rRNA genes amplified by PCR from extracted DNA from food samples has been proven to be a robust method of detecting foodborne pathogens, and >100 studies have been published [14,15,16,17,18]. The limitation of 16S rDNA gene sequencing obtained from PCR amplicons of extracted DNA from the microbial community (DNA-based) is the inability to differentiate live (dormant cells as well as growing or non-growing metabolically active cells) and dead cells [19]. In comparison, 16S ribosomal (rRNA) sequencing from RT-PCR amplicons (RNA-based) can target live microbial cells as both dormant and metabolically active cells producing rRNA [19]. Although there are different cell stains to differentiate between living and dead cells, such as the fluorescent dyes propidium iodide and SYTO 9 applied to foodborne pathogens [20], these microscopy-based approaches are generalist and cannot determine and identify which microbial species are active or dead. Other proposed strategies include the use of ethidium or propidium monoazide in combination with PCR or qPCR [21]. Here, we combined both culture-based methods and molecular approaches by means of high-throughput Illumina sequencing of 16S rRNA amplicons (DNA- and RNA-based) to study the bacterial community and to discriminate those active bacteria in RTE salads distributed by three of the most likely important and leading Spanish wholesale fruit and vegetable companies that also export to other European countries. Thus, these RTE products are consumed daily by hundreds of thousands of consumers. In Spain alone, according to National Federation of Fruit and Vegetable Producer and Exporter (Fepex), these three leading companies produced a total of ≈100,000 MT of different kinds of RTE vegetables in 2008, and a minor fraction (around ≤10%) of this production was exported to other European countries. Overall, data indicate a good microbiological quality in the studied RTE salad samples.

## 2. Materials and Methods

### 2.1. Sampling and Processing

Three different commercial brands of RTE packed salads were analyzed in the present study (Figure 1). These salads contained radicchio (*Cichorium intybus* var. *foliosum*), arugula (*Eruca sativa*), and lamb´s lettuce (*Valerianella locusta*). In all cases, RTE salads were sampled (triplicate) one day before the expiry date (01/17/2018) and processed for culture and nucleic acid extractions within the same day of sampling. RTE salads were purchased in two widely known Spanish supermarkets and immediately transported to the laboratory at 4 °C in less than 20 min. In the laboratory, according to EN ISO 6887-1, a total of 25 g with equal representation of the three leafy vegetables of each sample was added to 225 mL of buffered peptone water (Neogen Food Safety, Lansing, MI, USA) in Stomacher™ blender bags in sterile conditions and homogenized in a shaker (Labotron, Infors AG, Bottingen, Switzerland) at 400 rpm for 3 min. The homogenized food was then further processed for culture and molecular approaches. 

### 2.2. Microbial Cultures

In this study, total aerobic viable microorganisms (TVC), total coliforms, *Escherichia coli*, and *Salmonella* spp. were assessed. TVC were determined by spread-plating on Plate Count Agar (Oxoid) incubated at 30 °C for 48 h. For *E. coli*, we employed the RAPID’E.coli 2 Medium (BIO-RAD, Hercules, California, CA, USA) at 37 °C for 24 h. This medium is a selective chromogenic agar used for direct enumeration, without confirmation, of colonies of *Escherichia coli* and other coliforms in food products, and it has been certified and validated by AOAC according to the ISO 16140 standard. For *Salmonella*, the chromogenic RAPID’Salmonella medium (BIO-RAD, Hercules, California, CA, USA) certified and validated by the AFNOR certification according to the ISO 16140 standard was used at 37 °C for 48 h. In both vases, either for *Salmonella* and *E. coli*, we strictly followed the protocol according to the validated ISO 16140 standard.

### 2.3. Nucleic Acid Extraction and RNA Purification

From each sample, a total of 50 mL of homogenized food suspension was used for nucleic acid extractions. In order to remove large eukaryotic cells from vegetables, the suspension was pre-filtered through 5 µm sterile membrane filters (Millex-SV 5.0 µm, Merck Millipore, Madrid, España). As shown in the results, this pre-filtration did not remove all eukaryotic cells from vegetable tissues. The aim of this step was to increase the ratio of extracted microbial DNA and RNA. The suspension was then centrifuged at 6000× *g* for 20 min at 4 °C to collect the bacterial cell pellet. Nucleic acid (DNA and RNA) extraction were performed with the kit DNeasy® Blood and Tissue (ref. 69504, Qiagen, Düsseldorf, Germany) according to the manufacturer´s protocol detailed specifically for Gram-positive and -negative bacteria. Nucleic acids were stored at −80 °C until use. Concentration of DNA was measured by fluorometry with the Qubit HS dsDNA assay kit™ (ref. Q32851, Invitrogen, Carlsbad, California, CA, USA) in a Qubit 2.0 Fluorometer (Invitrogen. Carlsbad, California, CA, USA).

To purify the RNA from samples, an aliquot of the nucleic acid extraction was subjected to a rigorous DNase treatment with the TURBO DNasa™ Kit according to the manufacturer´s protocol (ref. AM1907, Invitrogen, Carlsbad, California, CA, USA). In brief, a total of 2 µL of TURBO DNase (2 U/µL) and 0.1 volume of 10× TURBO DNase buffer was added to 50 µL of nucleic acid sample that was incubated at 37 °C for 1 h. Afterward, 5 µL of inactivation DNase reagent was added, incubated at room temperature for 5 min, and centrifuged at 10,000× *g* for 1.5 min. The supernatant containing the RNA was transferred to a new tube and stored at −80 °C. RNA quantification was performed with the Qubit™ RNA HS assay kit in a Qubit 2.0 Fluorometer (Invitrogen, Carlsbad, California, CA, USA). To check whether the DNA was digested, the untreated and DNase-treated aliquot samples were both run in an electrophoresis gel in buffer TAE 1× with agarose 1% *w*/*v* (LE, Bio-rad, Hercules, California, CA, USA) treated with 0.1% *v*/*v* of diethyl pyrocarbonate to avoid RNA degradation. Electrophoresis confirmed that DNA was indeed digested and not present in the purified RNA aliquot. Furthermore, as described below, another quality control to ensure that recalcitrant undigested small DNA fragments were not present in the RNA, a purified aliquot was created based on the PCR of the 16S rRNA genes. In this PCR control, as a template, we used the RNA purified aliquot (see PCR conditions below). If undigested 16S rRNA gene fragments are present after the rigorous DNase treatment, then PCR amplicons would be obtained. As expected, no PCR amplification was obtained.

### 2.4. PCR and RT-PCR of 16S rRNA/rDNA and Illumina Amplicon Sequencing

The variable region of V3–V4 of the 16S rRNA/DNA was amplified by PCR and RT-PCR. This gene region is commonly used for high-throughput Illumina sequencing [16,22]. For the PCR, we used the primers 341 (CCTACGGGNGGCWGCAG) and 806 (GACTACHVGGGTATCTAATCC) following the PCR conditions of the standard protocol of Illumina (16S Metagenomic Sequencing Library Preparation). The expected size of PCR amplicons was checked in a conventional electrophoresis gel. 

The RT-PCR with the purified RNA aliquots was carried out in two steps. First, cDNA synthesis was done with the SuperScript IV First-Strand cDNA Synthesis Reaction (ref. 18091050; Invitrogen, Carlsbad, California, CA, USA) following the manufacturer´s recommendations with the same specific 16S rRNA gene primers used above. A total of 7.5 ng of RNA template was used from each sample for cDNA. PCR of 16S rRNA was then performed as above with the cDNA template following the PCR conditions and reagents recommended by Illumina (16S Metagenomic Sequencing Library Preparation). As stated above, PCR control amplifications were not obtained when purified RNA was used as a template, indicating that recalcitrant undigested 16S rRNA genes were not present in RNA and cDNA samples.

PCR and RT-PCR amplicons were sequenced in a MiSeq sequencer at the FISABIO Genomics Centre (Valencia, Spain) with 300 × 2 pair-end reads (15 Gb) and the MiSeq Reagent kit v3 according to the manufacturer´s protocol. Reads were quality assessed and trimmed with the Prinseq-lite program [23] applying the following parameters: “-min_length: 50, -trim_qual_right: 30, -trim_qual_type: mean, trim_qual_window: 20”. Reads R1 and R2 from Illumina sequencing were joined using FLASH program applying default parameters [24]. Afterward, joined reads were analyzed with Qiime2 (version 2018.11) software pipeline [25]. First, read sequencing data were quality-filtered again using “q-score-joined” with default parameters, chimeric sequences were eliminated, and representative sequences of each sample were retained and then assigned to taxa using SILVA 99% full-length trained classifier. Beta diversity was calculated using Euclidean distance and represented in a principal coordinate analysis (PCoA). Those operational taxonomic units (OTUs) classified in Qiime 2 as *Yersinia* spp. and other opportunistic pathogens were later manually checked in RDP and SILVA databases [26,27]. Furthermore, a 1000 bootstrapped phylogenetic trees of 16S rRNA gene sequences of *Yersinia* spp. based on neighbor-joining (NJ) and maximum likelihood (ML) models were calculated with the Geneious bioinformatic package [28]. First, sequences were aligned using the bioinformatic tool available in the SILVA database, manually checked, and then imported into Geneious. An ML bootstrapped tree (×1000) was calculated using the nucleotides substitution model HKY85 and option PHYML, while for the NJ tree (bootstrap of 1000), the Jukes–Cantor substitution nucleotide model was used. *Caulobacter vibrioides* and *Salmonella enterica* were used as outgroups. 16S rDNA/RNA reads obtained in this study can be downloaded from the data repository according to the guidelines of the journal. 

### 2.5. Real-Time PCR of Ail and FoxA Genes of Yersinia Enterocolitica

Real-time PCR of gene markers of *Y. enterocolitica* pathogenic strains was performed as previously described [29] with the exception that SYBR Green I dye was used instead of Taqman probes for monitoring the amplification. 

## 3. Results

### 3.1. Microbial Cultures

All RTE salads used in this study were sampled in supermarkets the day before the expiration day and processed for culture and molecular analyses within the same day. A total of three different RTE salad brands were studied (Figure 1, Brands S1, S2, and S3). From the same RTE packed salad, one fraction was used for culture and the other one for molecular methods. For the former, total aerobic viable microorganisms (TVCs), fecal coliforms, *Escherichia coli*, and *Salmonella* spp. were assessed following validated standards (see methods for details). TVCs in all analyzed samples were in the range of 2–4 × 10^6^ bacteria/mL and were quite homogeneous between different samples and brands. In all analyzed samples and brands, *Salmonella*, *E. coli*, and fecal coliforms (Figure 1) were not detected, indicating good agricultural practices and management of vegetables.

### 3.2. High-Throughput Sequencing of 16S rRNA

From each one of the analyzed brands, since culture-based data pointed to very similar results, one replicate sample from each was used for high-throughput 16S rDNA/rRNA sequencing of amplicons obtained by PCR and RT-PCR from extracted DNA and RNA (see Table 1 and Figure 2). The sequencing data obtained from 16S rRNA amplicons by PCR targeted the entire bacterial community—regardless of whether bacteria were dead or alive—while sequencing results obtained from 16S rRNA amplicons by RT-PCR from cDNA provided information about the active part of the bacterial community present in the analyzed RTE salads. For this, a rigorous and strict DNase treatment was applied (see Methods) to ensure that 16S rDNA genes were fully degraded and not present in the purified RNA (Figure 2). PCR controls carried out after DNase treatment indicated that recalcitrant 16S rDNA gene fragments to DNase were not present (Figure 2) in samples; thus, DNA was fully degraded. Afterward, 16S rRNA amplicons obtained from PCR and RT-PCR were Illumina sequenced, and between 60,000 and 100,000 quality-filtered reads were obtained (Table 1) from each sample. Qiime2 was used to analyze the data, and an operational taxonomic unit (OTU) threshold was set to 99% of nucleotide identity, with between 7201 and 12,848 OTUs obtained from sequenced samples. As expected, sequences and OTUs from the 16S rDNA/rRNA of chloroplasts and mitochondria from the leafy vegetables present in food samples were detected in samples but not considered for further analyses. The total number of classified genera detected in the analyzed samples ranged from 28 to 57 (Figure 3 and Table 1), although a significant fraction of bacteria remained unclassified at the genus level (5.4–36.9%). Samples from Brands S1 and S2 contained 28–37 different classified genera (Table 1), while Brand S3 had more genera (57 and 46 based on DNA and RNA-based approaches). As expected, in the 16S rRNA sequence data from RT-PCR amplicons (RNA-based), fewer classified genera were found in general, indicating that some of the bacteria observed in DNA lack ribosomal RNA and thus were totally inactive and/or possibly dead (Table 1). As shown in Figure 3, most bacteria present in the analyzed RTE salads belonged to the phylum Proteobacteria (90% of abundance). Other less abundant phyla (Actinobacteria and Bacteroidetes) were also detected. 16S rRNA sequencing data indicating the active bacterial fraction showed very similar results: a relative abundance of phyla (Figure 3). Overall, no major differences were observed at the phylum level between brands from both 16S rDNA and rRNA sequencing data. At the genus level, in all RTE salad brands, the most predominant and active bacteria (53–75% of frequency, Appendix A) were *Pseudomonas* spp. followed by other much less abundant bacteria, such as *Rahnella* or *Flavobacterium* (Figure 3). As shown in the principal component analysis (Figure 4), which takes into account the diversity and abundance of taxa, Brands S1 and S2 had a more similar microbial community structure and active bacteria (Figure 3), while Brand S3 differed significantly (Figure 4), although for those most abundant bacteria, the species composition was similar between brands (Table 1). 

### 3.3. Seeking Foodborne Pathogens in the Molecular Data

We sought to investigate whether 16S rDNA/rRNA sequences of well-known foodborne pathogens were present and active in the sequencing data from the analyzed RTE salads. In good agreement with culture-based methods, molecular data and taxonomic assignment with Qiime 2 confirmed and corroborated that no OTUs of *Salmonella* and *Escherichia coli* were found in samples. The same negative results were obtained for other pathogens, such as the psychrophilic *L. monocytogenes*. According to the Spanish and European legislation (e.g. CE n° 1441/2007), this pathogen has to be monitored in RTE food by culture methods. Among all known pathogenic bacteria, in one of the RTE salads (Brand S2), we detected a rare OTU (relative abundance 0.0015–0.003%, Appendix A) that was found in both DNA- and RNA-based approaches and that Qiime 2, RDP, and SILVA programs (see methods for details) classified as *Yersinia* spp. For instance, in BLASTn, some of the closest hits were related to *Y. enterocolitica*. Thus, it was paramount to unveil whether OUT was a potential pathogenic strain of *Y. enterocolitica*. However, a fine phylogenetic analysis by neighbor-joining and maximum likelihood models of this *Yersinia* OTU confirmed that it was related to non-pathogenic *Yersinia* strains (Figure 5). In line with these results, common virulent gene markers of *Y. enterocolitica* strains named as ail and foxA genes [29], could not be detected by qPCR from any of the samples.

Although much less disquieting a priori, we detected a few OTUs related to the opportunistic pathogens *Aeromonas hydrophyla* and *Rahnella aquatilis*. For the former, it was only detected in the Brand S2 from both DNA and RNA, albeit at an extremely low frequency (0.002%), and was assigned to *A. hydrophyla* strain ATCC7966. Regarding the enterobacterium *R. aquatilis*, several OTUs assigned to this species were detected in all analyzed samples and in brands at much higher relative frequency (1.3–7.73%) from both RNA and DNA sequencing data, indicating that this bacterial species was present and metabolically active. 

## 4. Discussion and Conclusions

In the present study, we have addressed the microbiological quality of three of the most distributed and consumed RTE salad brands in Spain that are also exported to other European countries. RTE vegetables are well-known to be a vehicle for foodborne bacterial pathogens, and in several cases have led to widespread and very serious outbreaks of foodborne illness, such as that of the *E. coli* O104:H4 associated with sprouts [30]. Poorly treated wastewater used in agriculture for watering might contain high numbers of enterobacteria, and this is one of the main factors contributing to the surface contamination of leafy vegetables in addition to the industrial processing of RTE products [1,31]. Soil microorganisms can also be transferred to the surface of leaves, so RTE salads can contain an important proportion of common soil taxa and microorganisms common in the endosphere and rizosphere of the plant [32]. Obviously, each plant harbors its own specific bacterial community that, to some degree, can be also shared by different vegetable species [33]. Indeed, for instance, *Pseudomonas* spp. is a typical genus found in soils and associated with plants [34] and in some cases is a common plant pathogen [35]. For other genera detected in our study (Figure 3), such as *Serratia* or *Pantoea*, there are multiples studies showing their ubiquity in plants (e.g. leave) and soils [36,37]. Thus, considering that leaves present in commercial RTE salads are of course not sterile, it is not surprising to find these microbes in our study.

Specifically, in our study, the bacterial communities of the studied RTE packed salads were clearly dominated by *Pseudomonas* spp. that were metabolically active. Previously, a study employing a DNA array-based method that targets 16S ribosomal DNA (rDNA) in packed, RTE vegetable salad showed that *Pseudomonas* spp. dominated salad batches containing either Norwegian or Spanish lettuce, before storage and after storage at 4 °C [34]. Here, we have employed in parallel culture and culture-independent methods to study the bacterial diversity and activity in RTE salads. In Spain, as in other European countries under the same regulation (CE n° 1441/2007 and CE n° 2073/2005), the study of *E. coli*, *Salmonella* and *L. monocytogenes* in RTE salads is mandatory. Both culture and molecular methods confirmed the absence of these foodborne pathogens, and no incongruences were obtained by these contrasting approaches. In line with our results, a culture-based survey conducted in 2008–2009 with RTE vegetable salads sampled in Spanish school canteens indicated the absence of bacterial pathogens in the analyzed samples. However, as stated in that study, visits and sampling were previously arranged, although food handlers were not informed about the exact day of the visit. In our study, samples were acquired as “typical” consumers at well-known Spanish supermarkets without any previous notification. In another culture survey of RTE lettuce salads (*n* = 142) from the Swiss market, *E. coli* was only found in five lettuce samples, while *Salmonella* spp. was not detected in any of the analyzed samples [38]. 

It is unquestionable that validated culture-dependent techniques are the “gold standard” (e.g., ISO standards) in food microbiology, but it is crystal clear that next-generation sequencing techniques are becoming the revolution in food microbiology [12,32]. In our study, we employed the sequencing of 16S rRNA that addresses an important question in food microbiology: Which part of the bacterial community is active in a food sample? Sequencing data from PCR amplicons of 16S rDNA gene cannot determine whether these bacteria are actually transcribing and thus contain ribosomal RNA (rRNA), which is an indicator of metabolically active cells. The cell’s total RNA pool is mainly composed of rRNA (82–90%), and total RNA and rRNA content correlates well with growth rate [39]. The analysis of 16S rRNA as a biomarker for metabolic activity has been frequently employed to identify the active bacterial fraction in a sample [19,39]. Bacteria might exist in nature in a range of different metabolic stages, such as dormant, active, and growing; in food microbiology, it is important to ascertain whether the detected bacteria are potentially active. There are more than 100 studies that have used rRNA and other biomarker genes for identifying active microbes (see list in [39]) including those from food samples [40]. Among the different culture-independent techniques to detect foodborne pathogens, PCR of 16S rDNA and sequencing on Illumina Miseq is currently the most commonly applied technique [12,32]. The advantage of these molecular methods is the potential to overcome the inherent sensitivity limitation of culture-based approaches since, in the same workflow (DNA extraction, PCR and high-throughput sequencing of 16S rRNA gene amplicons), we can detect and describe the whole bacterial community present in a food sample. An important question in food microbiology is to know the detection limit of the employed technique. Here, we used a well standardized technique that has been extensively proven to be very sensitive and robust. For instance, in a recent paper in which food samples were spiked with *E. coli* (10^1^–10^6^ cells/mL) and monitored using our PCR-based methodology applied for RTE salads (PCR of Region V3 and V4 of the 16S rRNA gene combined with Illumina Miseq sequencing), the authors demonstrated that it could be detected in all samples, as low as 101 cells/mL [41]. Another 16S rRNA-PCR-based study on *Salmonella* cells artificially contaminated in food samples indicated that the detection limit was N × 100 cells per assay [42]. Similarly, in *L. monocytogenes* inoculated in food samples, the PCR-based method was as low as 4–40 CFU [43].

Hygiene is a critical step in the farm-to-fork continuum, and all production/processing stages are designed to maintain sanitary standards of food manufacture [32]. In an ideal scenario of the farm-to-fork continuum, a total absence of foodborne pathogens and opportunistic bacteria is obviously desired. In our study as stated above, common foodborne bacterial pathogens were not found. However, one opportunistic pathogen, *R. aquatilis*, has been detected at considerable frequency. This bacterium has been detected in various infections, such as bacteremia from renal infection and respiratory infection, but in most cases in immunocompromised adults and organ transplant recipients [44]. To date, there has been no report on the foodborne transmission of this psychrotrophic coliform. In contrast, it has been described that *R. aquatilis* is involved in spoilage of refrigerated food (e.g. meat) and widely detected in other RTE salad samples, as shown here and in other vegetables and fruits [44,45,46,47,48,49]. For instance, in one survey on minimally processed vegetables, the dominating bacterial population during low temperature storage mainly comprised species belonging to the Pseudomonadaceae and Enterobacteriaceae, especially *Erwinia herbicola* and *R. aquatilis* [49]. 

We can conclude, based on our molecular and culture data from a random sampling of three of the most important fruit and vegetable producer companies, that there are no objective reasons to raise any major health concerns in the widely distributed and consumed RTE salads. Data indicate that these three RTE salad brands widely consumed in Spain and exported to other European countries have a good microbiological quality and a high standard in the processing of RTE vegetables, although a lower cell number of the opportunistic pathogen and food spoiler *R. aquatilis* would be desired. Furthermore, molecular and culture data are coherent. Lastly, the sequencing of 16S rRNA and rDNA provides a more complete and robust view of the bacterial diversity and the active bacterial fraction present in food samples.

## Figures and Tables

**Figure 1 foods-08-00480-f001:**
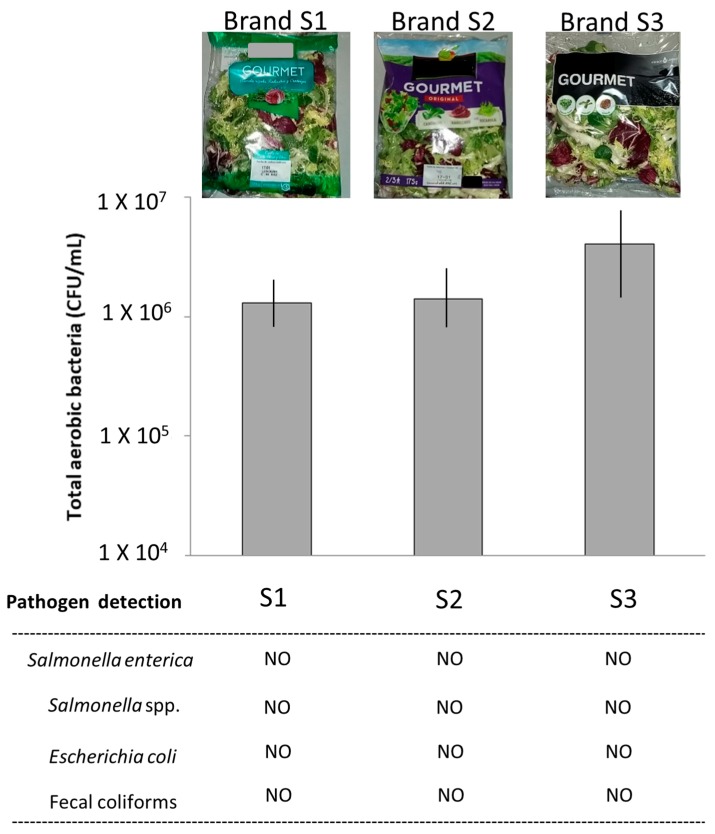
Microbial cultures and counts in the analyzed RTE salads. Image of the three different RTE brand samples analyzed in this study (top). From left to right, sample 1 (S1), sample 2 (S2), and sample 3 (S3). Total viable aerobic bacteria (CFU/mL) are shown for each one of the studied RTE salad brands. A total of three RTE salad brands were studied, named S1, S2, and S3. Triplicate samples were analyzed from each RTE salad brand (standard deviation of data is shown in bar chart). Detection of *Salmonella, E. coli*, and fecal coliforms is indicated. Cultures were performed according to validated ISO standards with chromogenic media (see methods for details).

**Figure 2 foods-08-00480-f002:**
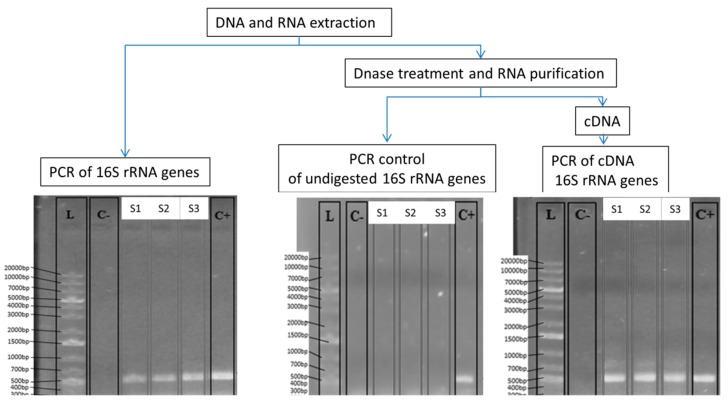
Workflow of the molecular approach (DNA- and RNA-based) used to characterize the bacterial community and the active bacterial fraction in the studied RTE salads. Electrophoresis shows the PCR and RT-PCR results of the 16S rDNA/rRNA. PCR control of undigested 16S rDNA genes is shown. C+: positive control of PCR and RT-PCR; C-: negative control of PCR and RT-PCR; L: ladder indicating the size (bp). Lanes marked as S1, S2, and S3 correspond to the analyzed RTE samples.

**Figure 3 foods-08-00480-f003:**
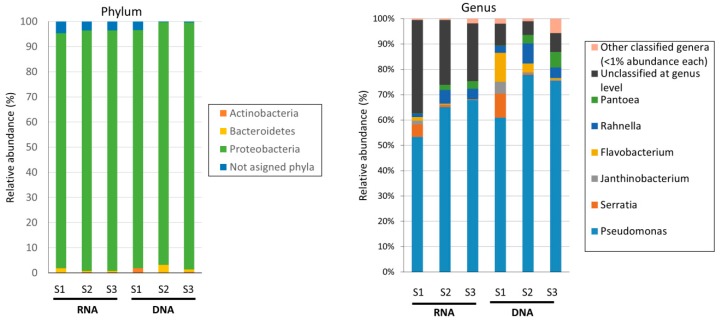
High-throughput sequencing data of 16S rDNA and rRNA from RTE salads. Taxonomic assignment of reads at the phylum and genus level is shown. From each RTE salad brand, taxonomic assignment data from total bacterial cells (dormant, dead, and active cells containing 16S rDNA genes) and metabolically active bacteria (cells containing ribosomal RNA, 16S rRNA) is indicated as “DNA” and “RNA”, respectively. Those low abundant genera that showed <1% of relative abundance were grouped within the category of “Other classified genera.” Relative abundance for each bacterial taxon is referred to as the total of bacterial reads obtained from sequencing data, since chloroplast and mitochondria were also retrieved but not considered for the abundance analysis.

**Figure 4 foods-08-00480-f004:**
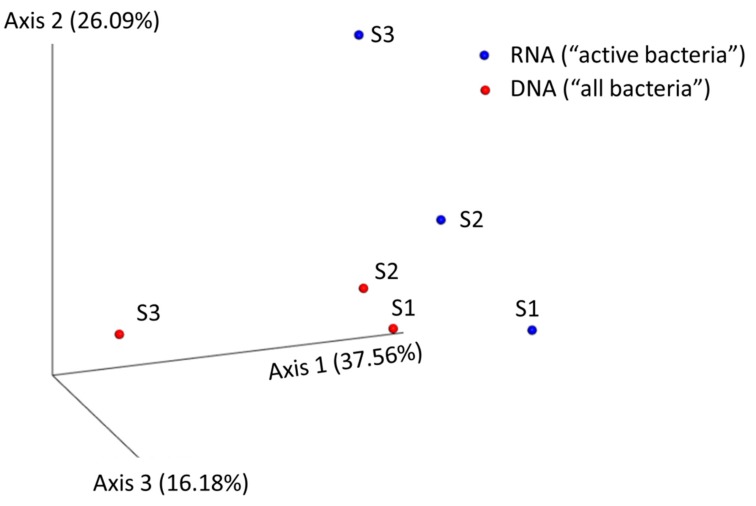
Principal component analysis (PcoA) of bacterial community structure at the genus level from the analysed RTE salads. The PcoA was performed with “qiime tool collapse” at the desired taxa level using the Euclidean metric to calculate the matrix distance. Samples from total bacterial cells (dormant, dead, and active cells containing 16S rDNA genes) and metabolically active bacteria (cells containing ribosomal RNA, 16S rRNA) are depicted in red and blue colors, respectively. Two close samples in the PcoA indicated that they had similar bacterial species composition and relative abundances of taxa. Although no major differences were observed regarding the abundant bacterial species composition, significant differences in relative abundances were observed (see Figure 2), especially for those low abundant species.

**Figure 5 foods-08-00480-f005:**
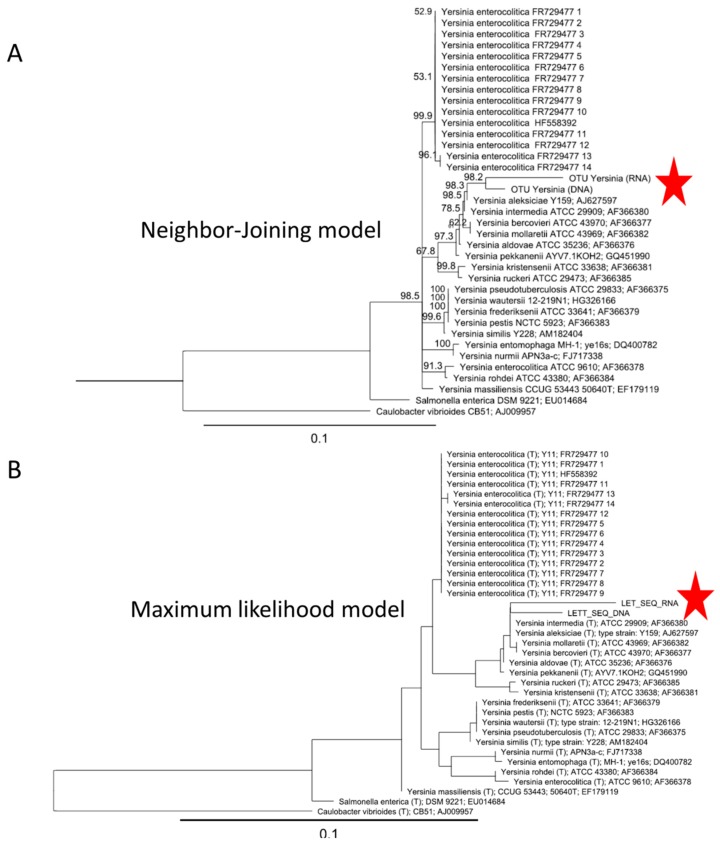
Phylogenetic tree of operational taxonomic units (OTUs) assigned to *Yersinia* spp. Reference type strains of all *Yersinia* species are included in the analysis. *Yersinia* OTUs obtained from the RTE salad samples are indicated with a red star. The consensus phylogenetic tree was calculated (×1000 replicates) based on neighbor-joining with a Jukes–Cantor substitution nucleotide model (**A**) and the maximum likelihood with the HK substitution nucleotide model **(B**). *Caulobacter vibrioides* and *Salmonella enterica* were used as outgroups.

**Table 1 foods-08-00480-t001:** Sequencing data after quality filtering and joining of forward and reverse reads, and the number of detected genera with Qiime. Abbreviations: sample 1 (S1), sample 2 (S2) and sample 3 (S3).

Sample	No. of Reads	Average Length of Joined Reads (bp)	SD of Length	Detected Genera by Qiime
**S1 (DNA)**	97,745	456.14	15.81	30
**S2 (DNA)**	78,252	454.41	16.17	32
**S3 (DNA)**	69,805	451.70	16.39	37
**S1 (RNA)**	86,122	238.47	166.95	28
**S2 (RNA)**	73,897	303.27	173.09	57
**S3 (RNA)**	58,758	338.65	167.04	46

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
