# Peer review of "High-Throughput 16S rRNA Sequencing to Assess Potentially Active Bacteria and Foodborne Pathogens: A Case Example in Ready-to-Eat Food"

_foods, 2019, doi:10.3390/foods8100480_

Round 1

Reviewer 1 Report

The manuscript is very well written and will be of great interest of the scientific community involved in food industry. I'm sure that it will get some attention of national media, because food safety is nowadays of a great interest of general public. It's very good news that the authors didn't find any foodborne pathogens by culture neither by 16 amplicon sequencing.

Miralles et al used a very smart method for distinguishing active and inactive cells. After reading the manuscript title, I was first thinking about the life and death cell staining for this purpose, but the RNA-based amplicon sequencing looks like an easy alternative and can be reproduced in any other lab. Maybe the authors can add a sentence about alternative methods for distinguishing active and inactive cells to the Introduction.

Only thing I'm missing is a paragraph about the microbiome of the plants in the Discussion. Authors wrote only one sentence on line 297-299. Readers who are not aware about the plant endosphere microbiome might be asking: "Where all these detected microbes come from?". In my opinion authors should compare the list of bacteria found in their study with other studies about plant leaf microbiome. It's just in order to explain that it's absolutely normal to find some bacteria in a plant material. Leaves in commercial salads can't be sterile, even if their surface is washed by bleach before packaging. This is just to be added to the Discussion.

In my opinion, the supplementary figures can be moved to the main text, and the supplementary tables are already described in the main text

For example, Figure S2 is very nice and it could go to the main text. Maybe Figure S1 and S2 can be combined into one figure with two panels and can go to the main text. The phylogenetic tree in the supplementary material can be combined with the tree in the main text, so it will have two panels.

In my opinion, the Table S1 can be removed, because the numeric description in lines 241-215 is sufficient. The same for the Table S2 - the description in the line 220 is sufficient. Table S3 can be also removed, because it is already visualized in the form of barplots in the Figure 2. This is just my suggestion, the final decision on whether to remove these tables or not is on the authors.

Overall, the article is very nicely written, its' scientifically sound, it involves interesting methods, the methods are described in sufficient detail and the outcomes are stated clearly.

Author Response

Dear reviewer and editor,

We appreciate very much the effort and time to review and edit this manuscript. We are glad that reviewers find that this manuscript is "very interesting, because it reflect the current era of culture independent pathogen detection, a very promising approach to detected pathogen in food(rev 2) and that is  "very well written and will be of great interest of the scientific community involved in food industry" and that consider that we have used "a very smart method for distinguishing active and inactive cells" and that is "scientifically sound, it involves interesting methods, the methods are described in sufficient detail and the outcomes are stated clearly" (rev. 1). As per reviewer´s suggestion, all supplementary material has been moved to main text. Thus the new version include a total of 5 main figures, with new panels coming from supplementary, and a new table 1. We have included all reviewer´s suggestion in this new version. We hope that now reviewers find this version suitable for publication.

Reply to Maybe the authors can add a sentence about alternative methods for distinguishing active and inactive cells to the Introduction. According to this referee, we have added a new paragraph in Introduction about other methods to distinguish active and inactive cells and now it reads "Although, there are different cell stains to differentiate between alive and death cells, such as the fluorescent dyes propidium iodide and SYTO 9 applied to foodborne pathogens (Zordan et al., 2009), these microscopy based approaches are generalist and cannot determine and identify what microbial species are active or death. Other proposed strategies are for instance the use of ethidium or propidium monoazide in combination with PCR or qPCR (Flekna et al., 2007).

Only thing I'm missing is a paragraph about the microbiome of the plants in the Discussion. Authors wrote only one sentence on line 297-299. Readers who are not aware about the plant endosphere microbiome might be asking: "Where all these detected microbes come from?". In my opinion authors should compare the list of bacteria found in their study with other studies about plant leaf microbiome. It's just in order to explain that it's absolutely normal to find some bacteria in a plant material. Leaves in commercial salads can't be sterile, even if their surface is washed by bleach before packaging. This is just to be added to the Discussion. We have extended our discussion regarding this topic as per suggestion of this referee. A new paragraph has been included.

In my opinion, the supplementary figures can be moved to the main text, and the supplementary tables are already described in the main text. For example, Figure S2 is very nice and it could go to the main text. Maybe Figure S1 and S2 can be combined into one figure with two panels and can go to the main text. The phylogenetic tree in the supplementary material can be combined with the tree in the main text, so it will have two panels. In my opinion, the Table S1 can be removed, because the numeric description in lines 241-215 is sufficient. The same for the Table S2 - the description in the line 220 is sufficient. Table S3 can be also removed, because it is already visualized in the form of barplots in the Figure 2. This is just my suggestion, the final decision on whether to remove these tables or not is on the authors. We have accepted all suggestions of this referee, and supplementary figures have been moved and merged with the main figures. Fig S1 has been integrated and merged with figure 1 into a single one. Fig S2 is now the main figure 2 and the phylogenetic tree in supplementary material has been moved into figure 4. Table S3 has been removed and we agree with this referee that this data is already depicted in the bar plot. Finally suplementary tables S1 and S2 have been merged into a single one and moved as maine table. We think is important to show the exact number of obtained raw sequences and number of detected genera for each one of the analyzed samples. Again, we thank this reviewer for her/his valuable suggestions. 

Reviewer 2 Report

The concept behind the study is very interesting, because it reflect the current era of culture independent pathogen detection. A very promising approach to detect pathogen in food.

The first concern is in the design of the study.  Not enough samples were tested to have a true statistical representation of the microbiome present in these RTE foods. 

A Second concern was that the study did not have a low or high detection or sensitivity parameter that would compensate for lower level bacteria that were present. 

Another concern is that bacteria such as E. coli, Salmonella, Listeria, prefer certain pre-enrichment conditions and this study did not really account for that in order the make the assumption that these organisms were not present or detectable.  The number of samples tested is not enough to draw such a conclusion.  

The approach and method is very promising for rRNA detection of pathogens.  As the author may have noticed it is difficult to truly isolate rRNA at a level that may reflect the true microbiome of foods.  Only the prominent organism, based on the results of this study are the plant associated organism were at high level.  

The study did not account for the variables that does exist in food testing.

Author Response

Dear reviewer and editor,

We appreciate very much the effort and time to review and edit this manuscript. We are glad that reviewers find that this manuscript is "very interesting, because it reflect the current era of culture independent pathogen detection, a very promising approach to detected pathogen in food(rev 2) and that is  "very well written and will be of great interest of the scientific community involved in food industry" and that consider that we have used "a very smart method for distinguishing active and inactive cells" and that is "scientifically sound, it involves interesting methods, the methods are described in sufficient detail and the outcomes are stated clearly" (rev. 1). As per reviewer´s suggestion, all supplementary material has been moved to main text. Thus the new version include a total of 5 main figures, with new panels coming from supplementary, and a new table 1. We have included all reviewer´s suggestion in this new version. We hope that now reviewers find this version suitable for publication.

Reply to Not enough samples were tested to have a true statistical representation of the microbiome present in these RTE foods. A Second concern was that the study did not have a low or high detection or sensitivity parameter that would compensate for lower level bacteria that were present. This current study does not intend to make a comprehensive analyses of the microbiome present in leaves of packages salads but the main objective is to prove that this methodology is robust and sensitive enough to detect and virtually target prokaryotes present in a food samples, such as the one studied here, vegerable salads. However, in terms of the general diversity found in the analyzed samples are not very different and agree in the most abundant and active genera. Thus, although, this pilot study had limited funding resources to sequence a small number of samples, even with that number, data agree quite well and point to Pseudomonas in all samples as the most abundant and active bacteria. In addition, in all samples, those oportunistic pathogens were detected in addition to non-pathogenic (a priori) Yersinia spp. Based on these results, we are in the position to ascertain that sequencing more samples, would not change anything the point of the story.

Regarding sensitivity, we would like to share with the editor and this reviewer, that the detection limit of PCR or RT-PCR-based methods, like the one used has been widely proved not only with food samples but with many other type of biological samples. For instance, first paper published back in 1992 by Wang and colleagues confirmed that "this method detected as few as 2 to 20 CFU of L. monocytogenes in pure cultures and as few as 4 to 40 CFU of L. monocytogenes in inoculated (10(8) CFU), diluted food samples". Another example has been published in Salmonella in foods by using PCR of 16S rRNA gene (Lin and Tsen, 1996), and I literally copy "When this PCR system was used for the detection of Salmonella cells artificially contaminated in food samples, results obtained were satisfactory. A detection limit of N× 100 cells per assay could be obtained." Just to cite another one, very recently, a year ago by Brandt and Albertsen (2018)  analyzing the detection limit of the region V3-V4 of 16S rRNA gene, precisely, the gene region and same primers used here in our study of salads. They stated that samples "spiked with Escherichia coli cells in different concentrations [101–106 cells/ml] could be detected in all sample". Thus, authors think that there is no need to be worried about the detection limit of the technique, in the order of units of bacteria per ml or gram of sample. See links to the above mentioned paper below

https://aem.asm.org/content/58/9/2827.short

https://www.ncbi.nlm.nih.gov/pmc/articles/PMC6137089/

https://onlinelibrary.wiley.com/doi/abs/10.1111/j.1365-2672.1996.tb03271.x

Reply to "Another concern is that bacteria such as E. coli, Salmonella, Listeria, prefer certain pre-enrichment conditions and this study did not really account for that". Precisely, we have followed international and standard norms to culture detected some of these pathogens, such as ISO 16140. Thus, there is no much room for debate within this topic. In the case of the uncultured approach, precisely, a strong point of this technique is that as PCR and sequencing based methods are so sensitive that can target up to 1 OTU or CFU/g in a sample, then there is no need to perform the enrichment, which is very valuable  precisely for saving time when the detection of foodborne pathogens in an outbreak is urgent. In any case, as discussed by this reviewer, this can be easily implemented in the method if needed with simply including the pre-enrichment media for each pathogen . However, bear in mind that each pathogen normally need a different enrichment media and conditions, which can be very tedious and in some cases depending on the pathogen can take up to 24-48h. However, with our PCR/RT-PCR method combined with next generation sequencing can virtually target any type of pathogen and prokaryote regardless the abundance and without the need of preparing several different enrichment media.

Reply to "The study did not account for the variables that does exist in food testing". We thank this comment but we are not sure what does he/she wanted to mean. We have intended to consider all possible real variables, and indeed, during our experimental design, before performing the experiment and sampling, we were advised by an expert food supervisors and sanitary spanish inspector to ensure that we followed strictly the protocol for food sampling. The sampling was done as real consumers at random picking different bags of RTE salads, brougth to the laboratory and immediately processed according to the international validated norm EN ISO 6887-1.